# Contributions of different host species to the natural transmission of severe fever with thrombocytopenia syndrome virus in China

Qu Cheng[1]*, Xinqiang Wang[1], Qi Li[1], Hailan Yu[1], Xiaolu Wang[1], Chenlong Lv[2], Junhua Tian[3], Banghua Chen[3], Zhihang Peng[4], Liqun Fang[2], Wei Liu[2], Yang Yang[5], Bethan V. Purse[6]

**1** Department of Epidemiology and Biostatistics, School of Public Health, Tongji Medical College, Huazhong University of Science and Technology, Wuhan, Hubei, China, **2** Institute of Microbiology and Epidemiology, Beijing, China, **3** Wuhan Center for Disease Control and Prevention, Wuhan, Hubei, China, **4** Chinese Center for Disease Control and Prevention, Beijing, China, **5** University of Georgia, Athens, Georgia, United States of America, **6** UK Center for Ecology & Hydrology, Wallingford, United Kingdom

* chengqu@hust.edu.cn

## Abstract

Severe fever with thrombocytopenia syndrome (SFTS) is a tick-borne emerging infectious disease with a reported mortality rate of up to 30% in hospitalized patients. The causative agent, SFTS virus (SFTSV) is maintained in nature through a transmission cycle involving animal hosts and ticks. Therefore, effective control of SFTS in nature environments necessitates a comprehensive understanding of the tick-host circulation patterns that sustain viral persistence. We developed and calibrated mathematical models using seroprevalence survey data collected across China to assess the relative contributions of diverse domestic and wildlife host species to transmission, their determinants, and the effectiveness of various interventions. Our analysis identified poultry, previously unrecognized, as the most important species across the majority of survey sites, followed by goat/sheep, cattle, and rodents. These rankings remained robust even when parameter values were perturbed or certain host species were omitted from the survey. Across all sites, increasing tick mortality rate $\mu_T$ and reducing transovarial transmission efficiency $\phi$ consistently ranked among the top five interventions that led to the most significant reduction in the overall $R_0$. Understanding the relative host contributions is crucial for developing interventions. Our simulation results indicated that halving the contact rate of the most important species with ticks could yield a 25-fold greater reduction in transmission intensity compared to halving that of the second most important species.

**Data availability statement:** All the code and data supporting the conclusions of this study are available at: https://github.com/qu-cheng/SFTS_host.

**Funding:** This work was supported by National Natural Science Foundation of China [grant number: 82204110 to QC], partially supported by National Natural Science Foundation of China [grant numbers: 82330103 to WL and 82320108018 to ZP] and National Key R&D Program of China [grant numbers: 2023YFC2306004 and 2022YFC2304000 to ZP]. The funders had no role in study design, data collection and analysis, decision to publish, or preparation of the manuscript.

**Competing interests:** The authors have declared that no competing interests exist.

## Author summary

Severe fever with thrombocytopenia syndrome (SFTS) is an emerging tick-borne disease characterized by rapid spread, a broad host range among vertebrates, high case fatality rate in humans, and potential for global dissemination due to the wide distribution of its main vector *Haemaphysalis longicornis*. Despite being identified nearly two decades ago, the specific contributions of different host species to its natural persistence remain poorly understood. In this study, we compiled published seroprevalence data across various host species from mulitple locations in China and calibrated mathematical models for each location. Our analyses identified poultry, previously unrecognized, as the most important species at the majority of locations, followed by goat/sheep, cattle, and rodents. Through simulations, we found that increasing tick mortality, reducing transovarial transmission efficiency, and lowering the contact rate between ticks and the most important host species are the most effective interventions for reducing local transmission intensity. Notably, halving the contact rate of the most important species with ticks could yield a 25-fold greater reduction in transmission intensity compared to halving that of the second most important species.

## Background

Severe fever with thrombocytopenia syndrome virus (SFTSV), also known as *Bandavirus dabieense* (formerly *Dabie bandavirus* and *Huaiyangshan banyangvirus*) [1], is the causative agent of SFTS, an emerging tick-borne zoonotic disease. The disease was first identified in 2009 in China, and has since attracted growing attention owing to its high case-fatality rate in humans [2–4], rapid spread in Asia [5–11], and potential for global dissemination due to the wide distribution of its main vector *Haemaphysalis longicornis* [12]. Though human-to-human transmission of SFTSV can occur via several routes [13,14], it is much less frequent than tick-human transmission [15], such that infected humans contribute minimally to the natural transmission of SFTSV. Consequently, the infection risk in humans is closely related to the transmission dynamics within non-human host populations. Therefore, understanding the natural transmission dynamics of SFTSV and the specific roles played by each host species is essential for developing targeted interventions to reduce the SFTS burden in humans.

However, previous studies on the role of different host species have predominantly focused on a single host species at a time in controlled environments, instead of assessing the role of multiple host species co-occurring simultaneously in natural settings. For example, field studies have highlighted the importance of goats due to their high seropositivity rates [16], while laboratory studies suggest that hedgehogs play a significant role given their wide geographic distribution, high tick burden, prolonged viremia period, and ability to sustain transstadial and non-systemic transmission of SFTSV [17]. Across tick-borne disease systems more

generally, vertebrate hosts can play a role in maintaining transmission in two main ways [18]. Transmission hosts, often small mammals, birds and reptiles, are competent hosts that play a direct role in transmission by replicating the pathogen and passing it onto new hosts when the tick feeds. Reproductive hosts are those that may not carry the pathogen themselves (perhaps mounting an immune response that prevents onwards transmission) but serve as a blood meal host for adult female ticks to reproduce and lay eggs and thus maintain and amplify the tick populations [18], elevating tick-host ratios. Both these types of hosts may also support non-systemic co-feeding transmission between infected and uninfected ticks feeding simultaneously on the same host [19]. Other ecological processes that influence geographical and seasonal variability in host roles in transmission and vector population dynamics could include host resource use dynamics, competition between host species, ratios of competent to non-competent hosts in the community and host predation of ticks [20,21].

The contribution of multiple host species to the natural transmission of pathogens in natural settings can be quantified by fitting mathematical models to species-specific prevalence rates collected from cross-sectional surveys in these settings. Their contributions can be represented by the species-level $R_{0i}$, which is the average number of secondary infections generated by a single case of that species. A larger $R_{0i}$ indicates a more significant role in local transmission. This approach has been employed to identify mallards and other dabbling ducks as reservoir host species for avian influenza A viruses [22], and rodents and bovines as key contributor to the transmission of *Schistosomiasis japonica* [23].

In this study, we systematically reviewed and compiled region-specific seroprevalence rates for various host species. Subsequently, we developed multi-host mathematical models to evaluate the relative contributions of various host species to the natural transmission of SFTSV. We aimed to rank the contributions of different host species by their $R_{0i}$s, elucidate their key determinants (e.g., species abundance, host preference of the ticks, and viremia duration in each host species), and identify effective interventions to reduce the transmission intensity of SFTSV.

## Methods

### Data collection

We systematically updated two existing reviews that covered literature published prior to December 31st, 2019 [24,25] to compile region-specific seroprevalence rates for various host species. We conducted an extensive search for more recent studies published between January 1st, 2020 and August 31st, 2023 on PubMed, Web of Science, Chinese National Knowledge Infrastructure database (CNKI), and Wanfang database. The key words used in the search were ("Severe fever with thrombocytopenia syndrome" OR "SFTS") AND ("seroprevalence" OR "seroepidemiology" OR "serum" OR "antibodies"), identical to those used in the previous reviews [24,25]. Only studies that reported original data and met the following criteria were included for further analysis: (1) seroprevalence rates were estimated from cross-sectional surveys, instead of from blood samples stored in animal hospitals; (2) included at least one species of livestock, poultry, pets, or small wild animals (e.g., rodents, birds), as large wild animals (e.g., wild boars, deer) are less abundant in China and thus contribute minimally to the overall prevalence; (3) host animals were sampled across the entire study area, not limited to areas near human cases; (4) seroprevalence rates were reported for at least three host species, with a minimum of ten individuals tested per species. All the included studies used a double-antigen sandwich enzyme-linked immunosorbent assay (ELISA) targeting the nucleocapsid protein of SFTSV to detect total antibodies, which may also cross-react with other less common bandaviruses, such as Guertu virus and Heartland virus [26,27]. For each host species, we extracted the number of individuals tested and the number of positive cases to calculate species-specific seroprevalence rates. Their confidence intervals were estimated using the Wilson score interval method, due to its robustness even when the estimated prevalence rate is 0 or 1 [28–30]. These confidence intervals were subsequently used to calibrate the mathematical models. Seroprevalence rates, rather than SFTSV RNA positivity rates, were used to calibrate the models because they were more frequently reported in the literature (i.e., only 3 out of 9 locations included in our study reported SFTSV RNA positivity rates) and were typically estimated with greater precision, due to their higher values (i.e.,

seroprevalence rates in poultry ranged from 0.84 to 57.09% in eligible studies, compared to 0 to 2.7% for RNA positivity rates) and consequently narrower confidence intervals.

## Development and calibration of the multi-host mathematical models

We developed and fitted a multi-host mathematical model separately for each survey to align the model outputs with the observed seroprevalence rates of all host species in that survey (Fig 1). The number of host species in the model was set to match the number of species surveyed. Domestic birds, including chicken, ducks, and geese, were collectively categorized as "poultry" in the model, while wild birds such as pigeons, pheasants, and turtledoves were grouped under "wild bird" (S1 Fig). Based on their disease status, the total population $N_i$ of each host species $i$ was partitioned into four subpopulations: the susceptible ($S_i$), exposed ($E_i$), infectious ($I_i$), and recovered ($R_i$) individuals. The tick population $N_T$ was partitioned into two subpopulations: the susceptible ($S_T$) and infectious ($I_T$) ticks. For simplicity, we did not differentiate between the various life stages of ticks (i.e., eggs, larva, nymphs and adults). We assumed closed and stationary populations for all host species and ticks, where their mortality rates equaled birth rates ($\mu_i$ for host species $i$ and $\mu_T$ for tick). SFTSV transmission between host animals and ticks occurs through tick bites. We assumed that the contact rates with ticks varying with host species ($\beta_i$ for host species $i$) due to the apparent host preferences of *H. longicornis* [31], and they were calibrated for each site separately based on the species-specific seroprevalence rates at the target site. Once bitten, individuals transition from $E_i$ to $I_i$ at a rate of $\gamma_i$, with $1/\gamma_i$ representing the average duration of the latent period in species $i$. An excessive SFTS-induced mortality rate of $\rho_i$ was assumed for host species $i$, while no excessive mortality rate was assumed for the ticks [32,33]. The durations of the infectious period $1/\sigma_i$ varied widely between species, ranging from one day in goats [16] to ten days in hedgehogs [17]. Transovarial transmission of SFTSV was included in the model by attributing a proportion $\phi$ of the offspring of infectious ticks directly to the $I_T$ subgroup. Notably, co-feeding transmission (i.e., the infection of susceptible ticks caused through feeding close to an infectious tick on the same host) was not included in the model, as it is rarely reported for SFTSV, although possible [17].

Since all eligible surveys were conducted in mainland China, where SFTSV has been circulating for an extended period [6], we assumed that the proportion of individuals in each subpopulation (i.e., susceptible, exposed, infectious, and recovered) had reached a steady state. Model parameters are shown in Table 1, while details of the model, including key assumptions, initial value for each state variable, and model equations are provided in S1 Text. The model calibration

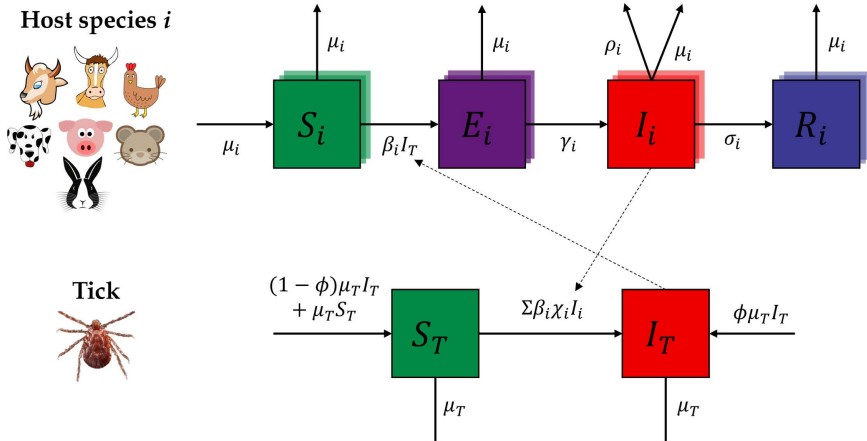

**Fig 1. Schematic of the multi-host mathematical model.** The symbols $S_i$, $E_i$, $I_i$, and $R_i$ represent the proportions of susceptible, exposed, infectious and recovered individuals of host species $i$, respectively. Similarly, $S_T$ and $I_T$ denote the proportions of susceptible and infectious ticks, respectively. The definition of each model parameter and its range in the calibration, the initial values of state variables, and model equations are provided in S1 Text.

**Table 1. Definitions of model parameters and their ranges for calibration.**

| Symbol | Description (units) | Value and notes | Reference |
|---|---|---|---|
| $1/\mu$ | Life span (days) | Goat/sheep: 438 (1.2 years)<br>Cattle: 949 (2.6 years)<br>Poultry: 318 (0.872 years)<br>Dog: 737 (2.02 years)<br>Pig: 274 (0.75 years)<br>Rodent: 365 (1 years)<br>Hedgehog: 1095 (3 years)<br>Weasel: 4745 (13 years)<br>Hare: 1095 (3 years)<br>Wild bird: 1095 (3 years)<br>Tick: 10–1095 (10 days to 3 years) | [36,37] |
| $\beta_i$ | Number of effective tick contacts per day (days$^{-1}$) | 0-2 for all species | |
| $1/\gamma_i$ | Duration of the latent periods (days) | 2-5 for goat/sheep, cattle, dog, and pig<br>0.5-3 for poultry and wild bird<br>1-3 for rodent, hedgehog, weasel and hare | [16,17,38–40] |
| $v_i = \frac{\rho_i}{\rho_i+\mu_i+\sigma_i}$ | Case fatality rate (unitless) | 0-0.1 for poultry and wild bird and 0-0.01 for all other species<br>We sampled $v_i$ and $\sigma_i$ in the calibration and used their values to calculate $\rho_i$, the excessive mortality rate caused by SFTSV infection | [16,17,38–40] |
| $1/\sigma_i$ | Duration of the infectious periods (days) | 0.5-4 for goat/sheep, cattle, dog, and pig<br>1-10 for poultry, weasel, hare, and wild bird<br>1-6 for rodent<br>9-11 for hedgehog | [16,17,38–40] |
| $\phi$ | Proportion of offspring born to infectious ticks that are infectious (unitless) | 0-1 | |
| $\chi_{max}$ | Relative abundance of the most abundant species to ticks ($\frac{N_{max}}{N_T}$, unitless) | 0-1<br>We sampled $\chi_{max}$ in the calibration and used it to estimate the relative abundance of species $i$ as $\chi_i = \frac{\chi_{max}N_i}{N_{max}}$ (S2 Text) | |

process is described in S2 Text. In brief, we randomly sampled parameters from the ranges defined in Table 1 until we accumulated 1,000 sets of parameters that generated simulated seroprevalence rates (equilibrium values of $R_i$s) within the 95% confidence intervals of their observed values for all surveyed species (referred to as *passes*) and resampled them using the sampling-importance resampling approach [34,35] to generate the posterior samples for estimating the overall and species-specific basic reproduction numbers.

## Estimation of the basic reproduction numbers

We used the next-generation matrix (NGM) method [41] to estimate the $R_0$ and $R_{0i}$s (S3 Text). To estimate $R_0$, the changes between pairs of subpopulations were decomposed into a transmission matrix $\boldsymbol{T}$ and a transition matrix $\boldsymbol{\Sigma}$, then the overall $R_0$ was estimated as the dominant eigenvalue of the next-generation matrix $\boldsymbol{K} = \boldsymbol{T\Sigma}^{-1}$. Species-level $R_{0i}$s were estimated in a similar way, while only the species of interest was kept in the matrices [42]. After simplification, the overall $R_0$ can be estimated as $R_0 = \frac{1}{2}(\phi + \sqrt{\phi^2 + \sum_{i=1}^{k} \frac{4\beta_i^2 \chi_i \gamma_i}{\mu_T(\mu_i+\gamma_i)(\mu_i+\rho_i+\sigma_i)}})$, while the species-level $R_{0i}$ as

$R_{0i} = \frac{1}{2}(\phi + \sqrt{\phi^2 + \frac{4\beta_i^2 \chi_i \gamma_i}{\mu_T(\mu_i+\gamma_i)(\mu_i+\rho_i+\sigma_i)}})$. Definitions of the parameters can be found in Table 1, while details about the

NGM can be found in S3 Text.

## Determinants of the species-level basic reproduction number

We assessed the importance of each transmission parameter on $R_{0i}$ through systematically perturbing parameter values and fitting a random forest regression to the results. Specifically, we first systematically perturbed each parameter

(Table 1) of the 1000 passes at a time by multiply it with values ranging from 0.1 to 10 (i.e., 0.1, 0.2, …, 0.9, 2, 3, …, 10), while keeping the other parameters at their original values. The perturbed parameter sets were then used to run simulations and the resulted passing parameter sets were used to re-estimate the $R_0$ and $R_{0i}$s. Next, we fitted a random forest regression model to the perturbed dataset, treating $R_{0i}$ as the dependent variable and the perturbed parameter values as the independent variables. The permutation importance for each parameter [43] was recorded and their uncertainties were obtained through bootstrapping.

### Optimal interventions for each location

Various interventions can be implemented to mitigate the spread of SFTSV, such as the use of essential oils to reduce the contact rate between host species and ticks ($\beta_i s$) due to repellent effects, the application of acaricides to increase tick mortality (increase $\mu_T$), culling to decrease the abundance of host species ($\chi_i s$), treatment of diseased hosts to shorten their infectious periods ($1/\sigma_i s$), and the development of genetic modification techniques to reduce the efficiency of transovarial transmission of SFTSV ($\phi$) in ticks [44]. In this analysis, we identified the five parameters among these that lead to the most substantial reduction in the overall $R_0$ as the most optimal interventions to reduce local transmission intensity.

### Impact of missing key host species in the survey

The seroprevalence surveys usually encompass only a limited number of species. To evaluate the impacts of omitting key host species from these surveys on the results, we fitted mathematical models to a subset of the surveyed species, systematically excluding one species at a time. We used data from Surveys 9 and 7 as examples, since they had the largest and second-largest number of host species, respectively. We excluded one species from the model structure at a time (Fig 1), recalibrated the model using the remaining survey data without this species, and re-estimated the $R_0$ and $R_{0i}$s. We then compared these recalculated values with those estimated with the complete dataset to assess the impact of missing each host species.

## Results

### Seroprevalence rates in host animals

A total of 47 studies were identified for full-text screening, consisting of 30 from two previous systematic reviews and 17 from our search for the most recent literature (S2 Fig). Of these, only eight studies met our inclusion criteria and were utilized to calibrate the mathematical models (S1 Table), all of which were conducted in mainland China (Fig 2A). Studies conducted in other countries either exclusively focused on wild animals [45,46] or estimated seroprevalence rates using stored blood samples from animal hospitals rather than through cross-sectional surveys [47]. We assigned unique IDs to the eligible surveys ordered by their publication years. If a single study covered multiple locations and reported results separately for each location, distinct identifiers were assigned accordingly; otherwise, the same ID was used. This process yielded nine surveys derived from eight studies. All these surveys were all conducted in regions with relatively high incidence rates of SFTS among humans [48], such as Shandong province, the border areas between Henan and Hubei provinces, and the border areas between Anhui and Jiangsu provinces. Some locations were surveyed more than once (i.e., Surveys 1 vs. 7 and 2 vs. 4, S1 Table), revealing similar species-specific seroprevalence rates over time (Fig 2B).

The eligible studies surveyed a total of 10,281 animals across 17 species, with goats/sheep, dogs, pigs, chickens, cattle, and rodents as the most frequently surveyed, while yellow weasels, hares, pheasants, rock pigeons, turtledoves, badgers, squirrels, and wild boars as the least often surveyed species (S1 Fig). Of these, 2,901 (28.2%) animals were positive for SFTSV total antibodies. Yellow weasels showed the highest pooled seroprevalence rate at 91.1%, (95% CI: 79.3-96.5%), followed by goats/sheep (63.5%, 60.9-66.0%), hares (63.0%, 51.5-73.2%), cattle (52.2%, 49.9-54.4%), and pheasants (42.9%, 21.4-67.4%). Species with fewer than ten individuals surveyed (i.e., squirrel, badger, and wild boar)

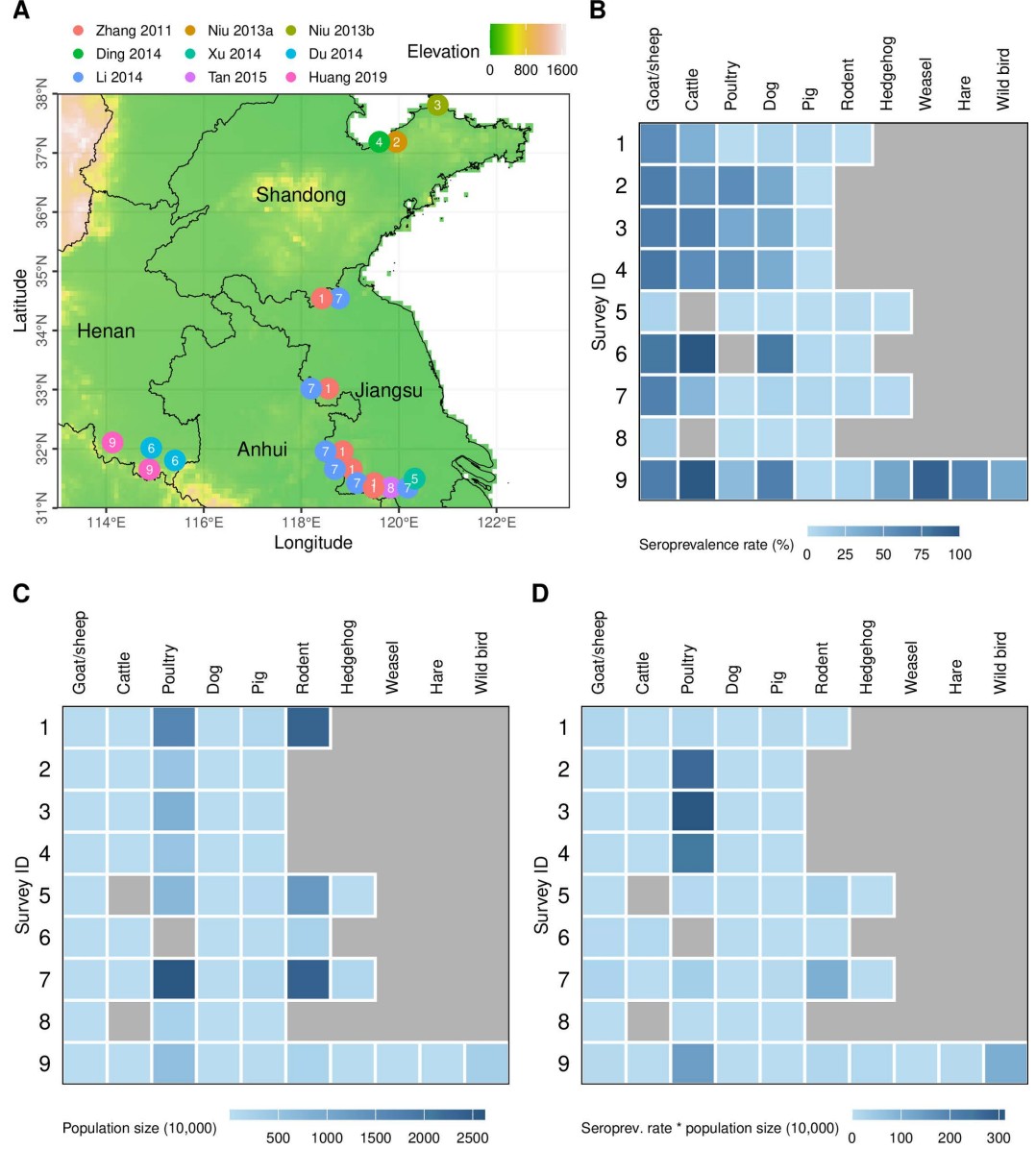

**Fig 2. Information of the included studies.** (A) Locations of eligible surveys. The dots for Surveys 1, 2, 4, and 7 were jittered to avoid overlap. (B) Seroprevalence rate for each species by survey ID. (C) Estimated abundance of each host species in the studied prefecture (See S2 Text for details on host abundance estimation). Only relative abundance was used in the model and it was assumed that the relative abundance of each host species in the surveyed locations matches that of the corresponding prefecture. (D) Estimated number of seropositive individuals for each species, calculated as the product of seroprevalence rate and host abundance. Survey IDs were assigned according to publication year. Provincial boundaries and elevation map were obtained from GADM (https://gadm.org/) and WorldClim (https://www.worldclim.org/), respectively.

were excluded from further analyses, leaving us 14 species aggregated into ten species groups for further analyses (Fig 2B). Comparison across surveys revealed higher seroprevalence rates in Surveys 6 and 9, while lower rates were observed in Surveys 5 and 8, suggesting varying transmission intensities across locations. Across animal species, average prevalence rates were consistently high in goats/sheep and cattle, consistently low in pigs and rodents, and varied significantly across locations for poultry and dogs (Fig 2B and S1 Table). As shown by the expressions for $R_0$ and $R_{0i}$, the

abundance of each host species critically influences local transmission intensity. While poultry and rodents exhibit the largest estimated population size (Text S2) across surveyed locations (Fig 2C), poultry contributed disproportionately more seropositive individuals due to their remarkably higher seroprevalence rates compared to rodents (Fig 2D).

### Overall $R_0$s

The median overall basic reproduction number $R_0$ varied across surveys, ranging from 1.02 (95% CI: 1.01-1.07) for Surveys 5 and 8 to 2.53 (95% CI: 1.42-4.20) for Survey 6 (Fig 3A). When compared with the other surveys, Survey 6 had a remarkably higher median value and a wider confidence interval for the overall $R_0$, which is likely due to the omission of poultry in the survey (see *Impact of missing key host species in the survey* of the *Results* and *Discussion* for details). Additionally, $R_0$s for locations surveyed multiple times in different years (i.e., Surveys 1 vs. 7, and 2 vs. 4) largely over-lapped, supporting the hypothesis that the transmission of SFTSV in mainland China has reached equilibrium. As the overall $R_0$ rose, the seroprevalence rates in goats/sheep, cattle, poultry, dogs, and hedgehogs increased rapidly, while those for pigs and rodents remained relatively stable (Fig 3B). This pattern implies that the seroprevalence rates of the former group were more strongly correlated with spatial variation in transmission intensity compared to the latter group, and can serve as sentinels for identifying regions with elevated SFTSV transmission intensity.

### Species-level $R_{0i}$s

Fig 4 shows the species-specific contributions to SFTSV transmission across various surveys. The most important host species varied across surveys, with poultry as the most important species (i.e., with the highest species-level $R_{0i}$) in five

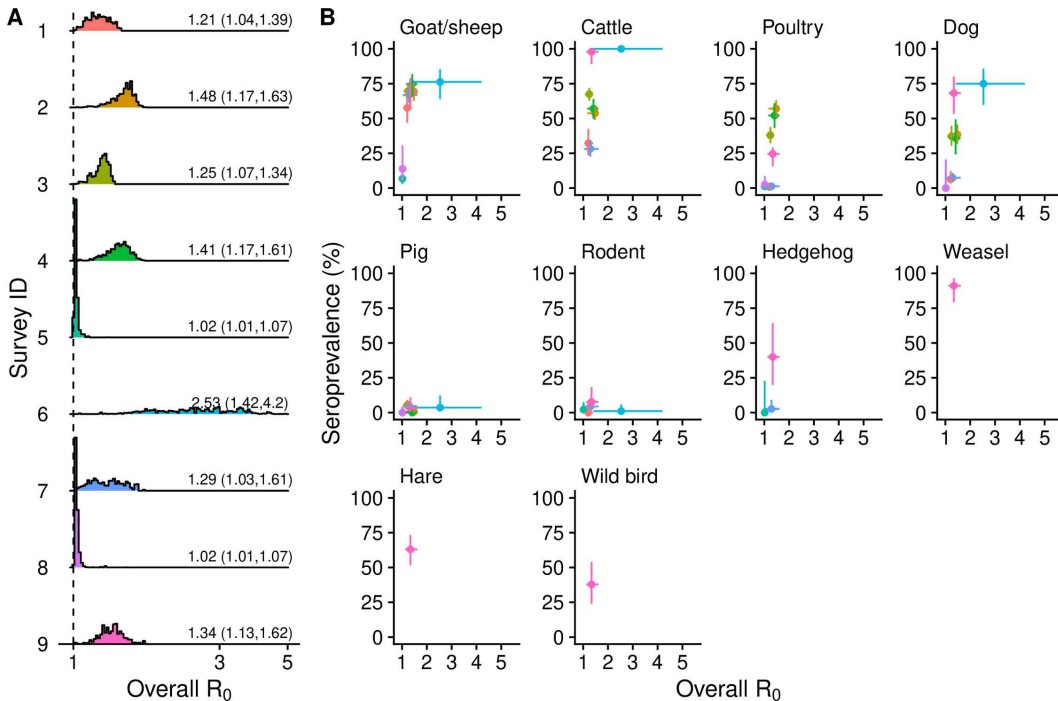

**Fig 3. Overall basic reproduction number.** (A) The posterior distribution of the overall $R_0$ for each survey. The vertical dashed line represents $R_0 = 1$ and the numbers on the right side of the plots indicate the median value and the 95% CI of the overall $R_0$ estimated from 10,000 posterior samples. (B) Relationship between the overall $R_0$ and the seroprevalence rates by species. Each dot represents a survey, with the same color scheme as in (A). The bars represent the 95% CIs.

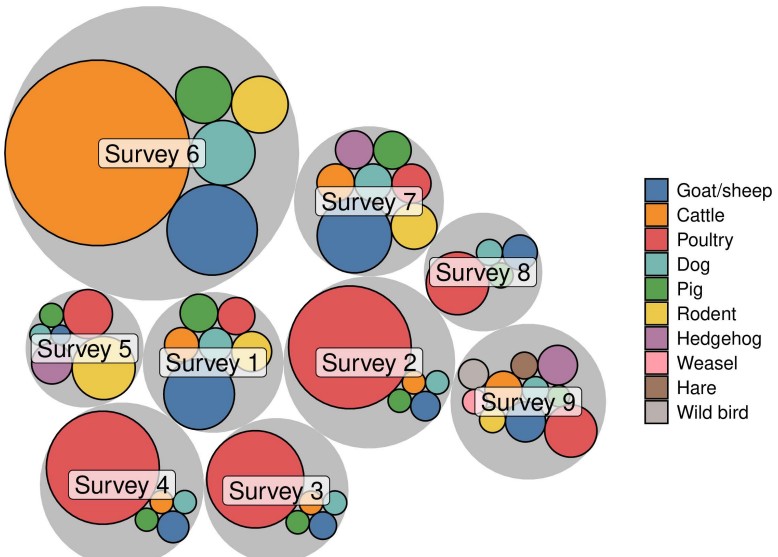

**Fig 4. Species-level $R_{0i}$s for each species in each survey.** The circles are colored by species. The radii of the gray circles are proportional to the median overall $R_0$s, while those of the colored circles are proportional to the species-level $R_{0i}$s.

surveys at four locations (Surveys 2, 3, 4, 8, and 9, with Surveys 2 and 4 conducted at the same location, S1 Table), goat/sheep in two surveys (1 and 7) at one location, and cattle (Survey 6) and rodent (Survey 5) each at one location. The contributions of dogs, pigs, hedgehogs, weasels, hares, and wild birds are limited. Detailed median values and 95% CIs of the overall $R_0$ and species-specific $R_{0i}$s for each survey are provided in S1 Table.

### Determinants of the species-level $R_{0i}$

The random forest model identified the contact rate with ticks $\beta_i$ and host abundance $\chi_i$ as the most important determinants of species-level $R_{0i}$, followed by the transovarial transmission efficiency ($\phi$), host mortality rate ($\mu_i$), host viremia duration ($1/\sigma_i$), and tick mortality ($\mu_T$) (S3 Fig). Notably, for each survey, although the exact values of $R_{0i}$s for different species were sensitive to these parameters, their relative rankings remained largely consistent across perturbations, indicating that our importance rankings of the species were robust (S4 Fig).

### Optimal interventions

Across all surveys, increasing tick mortality rate $\mu_T$ and reducing transovarial transmission efficiency $\phi$ consistently ranked among the top five interventions that led to the most significant reduction in the overall $R_0$ (S5 Fig). The remaining three interventions all focused on the most important species for each location (represented by the largest colored circles in Fig 4, also highlighted in bold in S1 Table), including reducing their contact rate with ticks $\beta_i$, abundance $\chi_i$, and viremia duration $1/\sigma_i$. These targeted interventions against the most important species resulted in more pronounced reductions in $R_0$ when these species had a higher relative contribution (i.e., a higher $R_{0i}$, S1 Table and S5 Fig). Targeting the second most important species was much less efficient. For instance, for Survey 2, reducing the abundance of poultry (the most important species as indicated by the highest $R_{0i}$ in S1 Table) to ten percent of its original value decreased the overall $R_0$ from 1.48 to 0.63, while reducing the abundance of goats/sheep, the second most important species, only marginally reduced the overall $R_0$ to 1.47.

## Impact of missing key host species in the survey

When recalibrating the model to the seroprevalence rates of the remaining species after systematically omitting one species at a time from Surveys 9 or 7 (represented by bars with different colors, Figs 5 and S6), the overall $R_0$ (Figs 5A and S6A) and species-level $R_{0i}$ (Figs 5B and S6B) remained largely stable, except when the species with the highest $R_{0i}$ was left out. In Survey 9, excluding poultry, the species with highest relative contribution but relatively low seroprevalence rate, led to remarkable increase in both $R_0$ and $R_{0i}$s. In contrast, in Survey 7, excluding goat/sheep, the species with both the highest relative contribution and the highest seroprevalence rate, resulted in a considerable decrease in both $R_0$ and $R_{0i}$s.

## Discussion

*H. longicornis* has been reported to feed on 77 vertebrate host species [12]. Understanding the roles played by these hosts in maintaining the natural transmission of SFTSV is critical for developing targeted interventions to reduce its transmission intensity. While previous laboratory study and animal seroprevalence survey attempted to address this question [17,49], their single-species focus limits their ability to capture the complexity of real-world transmission with multiple hosts. In this study, we employed multi-host mathematical models and the next-generation matrix approach to quantify the contribution of each host species to the natural transmission SFTSV. Our findings revealed that the most important host species varied across locations. Specifically, poultry was identified as the most important species for four out of seven locations examined, while goats/sheep, cattle and rodents each for one location. Key factors influencing host contributions included their abundance, contact rate with ticks, mortality rates, and viremia duration. The first three factors are location-specific, shaped by local ecological conditions, host community composition, and animal husbandry practices, while viremia duration is a trait tied to host immune responses but unmeasured for several host species with high prevalence rates, such as cattle, poultry and pigs. Intervention strategies that effectively increase tick mortality rate, blocking transovarial transmission of SFTSV in ticks, and reducing susceptibility and infectivity of the most important host species can reduce the overall $R_0$ to the largest extent, underscoring the importance of applying acaricide on the most important host species in each region.

   We highlighted the potentially critical role played by poultry in the natural transmission of SFTSV for the first time. Previous studies have emphasized the importance of goat/sheep due to their high seroprevalence rates [49] and hedgehogs because of their wide geographic distribution, high tick burden, long viremia duration, and low disease severity [17].

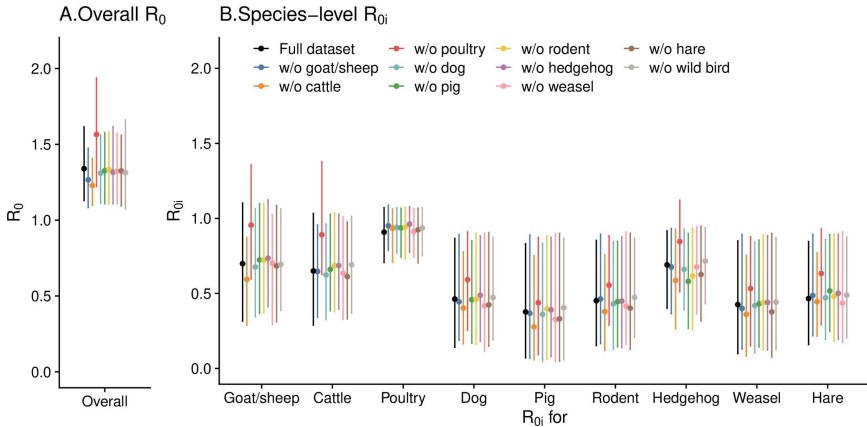

**Fig 5. (A) Overall $R_0$ and (B) species-level $R_{0i}$s when different species was left out from the calibration for Survey 9.** Black bars represent the values estimated using the full dataset, while the colored bars represent the values estimated when the seroprevalence rate for one host species was left out from the calibration.

However, our model results disagreed with these due to the very short viremia duration of goat/sheep and the relatively low abundance of hedgehogs compared to farm animals. Despite typically lower SFTSV seroprevalence rate in poultry compared to goats/sheep and cattle, their significantly higher abundance may lead to a more substantial contribution to the transmission dynamics. Notably, the seroprevalence rate of SFTSV in poultry varied widely across different locations (Fig 2B), which may be attributed to variations in poultry farming practices, as free-range poultry tends to exhibit higher seroprevalence rates than those in confinement [24], presumably due to their greater contact rates with tick questing habitats. Poultry was identified as the most important contributor for Surveys 2, 3, 4, 8, and 9, all of which reported high seroprevalence rates in poultry, except for Survey 8 where the seroprevalence rates were low for all species, possibly due to the low overall $R_0$ (Fig 3A).

Moreover, our results suggested that the most important host species for SFTSV transmission varied across regions, which was also observed for other infectious diseases. For instance, bovines were the most important species for the transmission of schistosomiasis japonica in the marshland region, whereas rodents predominated in hilly areas [23]. This spatial heterogeneity is likely influenced by local ecological context, including the relative abundance of host species, the role each host plays in driving tick population dynamics, and the contact frequency between ticks and host animals affected by resource-consumer dynamics, competition and predation, which are further shaped by animal husbandry practices, landscape and ecosystem context. These results align with a previous theoretical study, which suggests that the importance of a host species relies not only on its own characteristics, but also on the characteristics of other species within the community [50].

As a result, interventions should be tailored to target different host species at different locations based on their significance to local SFTSV transmission. For example, reducing the contact rate between goats/sheep and ticks to half of its original value for Survey 1, where they were identified as the most important species, resulted in a 21.5% reduction in the overall $R_0$, while implementing the same measure for Survey 2, where goats/sheep were the second most important species, led to only a 0.5% reduction. Conversely, targeting poultry, the most important species for Survey 2, could achieve a 42.3% reduction in the overall $R_0$. The results for other locations can be found in S1 Table. On average, halving the contact rate of the most important species with ticks resulted in a 25-fold greater reduction in overall $R_0$ when compared with halving that of the second most important speceis. Common intervention strategies include the application of acaricides and anti-tick vaccines on host animals, identification and treatment of the infectious hosts, and culling to reduce host animal abundances. Among these, the usage of acaricides might be the most effective, due to its dual function of increasing tick mortality rate and reducing the contact rate between ticks and hosts. To maximize cost-effectiveness, it is advisable to prioritize acaricide usage on the most important host species though uptake will depend on how these fit into local agricultural practices and economics.

Our findings have several limitations that should be considered when interpreting the results. Firstly, the development of the mathematical model and the selection of parameter ranges were based on the best available knowledge, which may not fully capture the real-world complexities. For instance, co-feeding transmission was not included, as it is understudied for SFTSV [17]. If co-feeding transmission does play an important role in the natural transmission, the importance ranking obtained in this study may change remarkably. Additionally, we assumed an infectious period of 1–10 days for poultry, which was only measured in doves [38], but not chicken and ducks. Future experimental infection studies are needed in poultry to better understand their clinical symptoms, duration of viremia, case-fatality rate, and antibody dynamics. We also assumed lifelong antibody persistence across host species based on observations of long-lasting antibodies in humans [51]; however, antibody half-lives remained unmeasured in non-human host species. Furthermore, repeated exposure to SFTSV in some host species may also sustain elevated antibody titers through immune boosting. These interspecies differences in antibody kinetics and exposure history may also contribute to the large variation in seroprevalence rates observed across host species. Secondly, our model assumed that equilibrium was achieved for each location. While comparisons of the results between Surveys 1 vs. 7 and 2 vs. 4 support this assumption, formally testing it for all locations were not feasible. Thirdly, our estimates are contingent upon results from previous seroprevalence surveys, which may leave out some key species. For example, cats were found to be able to transmit SFTSV [52] but were never included in the seroprevalence surveys. However, we assessed the influences of this

limitation by refitting the model to only part of the data collected by Surveys 9 or 7. The estimates remained robust, except when the most important species was excluded, which led to inaccurate estimates of overall $R_0$ and $R_{0i}$s for the remaining species. When a key species is left out from the model, to maintain similar seroprevalence rates for the remaining species, the estimated values for tick-related parameters, such as transovarial transmission efficiency $\phi$, tick mortality rate $\mu_T$, abundance-related parameter $\chi_{max}$, and species-specific biting rate $\beta_i$, were substantially affected, which further affected the overall $R_0$ and species-specific $R_{0i}$s. Fourthly, we assumed homogenous mixing between ticks and host species across life stages, which may not always hold true, since ticks feeding on a particular species might have a higher likelihood of re-feeding on the same species during their later life stages, because their habitats may largely overlap with the habitats of this species. Ignoring this heterogeneous mixing pattern could lead to underestimation of $R_{0i}$. At last, all the eligible surveys were conducted in China. Given the spatial variation in ecological context, such as tick habitat types, human agroecosystems, and wildlife abundance, key species might differ in other countries.

In conclusion, our findings indicate that the most important host species for the natural transmission of SFTSV vary by geographic locations. While poultry was identified as the predominant host across multiple locations, particularly where their seroprevalence rates were notably high, goat/sheep, cattle, and rodents also emerged as important hosts in certain areas. This spatial heterogeneity must be taken into account when developing intervention strategies. We recommend applying acaricides to the most important host species, as this approach can simultaneously reduce contact rates between hosts and ticks and eliminate ticks. To minimize potential bias in our results, future studies should prioritize investigating non-systemic transmission mechanisms between ticks and measuring key biological parameters of SFTSV, such as the duration of the infectious period for each host species.

## Supporting information

**S1 Text. Multi-host mathematical model.**
(DOCX)

**S2 Text. Model calibration.**
(DOCX)

**S3 Text. Estimation of the basic reproduction numbers.**
(DOCX)

**S1 Table. Basic information, estimated overall R$_0$, and species-level R$_{0i}$s of the included seroprevalence surveys.** The numbers in the parentheses are the 95% CI. The most important species for each survey are highlighted in bold.
(DOCX)

**S1 Fig. Seroprevalence rates of SFTSV in different animal species.** The dots represent the point estimates, while the error bars represent the 95% confidence interval determined with the Wilson score interval method. The sizes of the dots represent the total sample size aggregated across studies, while the colors of the dots represent the number of studies that surveyed each species.
(DOCX)

**S2 Fig. Flowchart of the study selection process.**
(DOCX)

**S3 Fig. Importance of each parameter in predicting the species-level R$_{0i}$.** The length of the bar represents the median permutation importance score from 100 repetitions of fitting, each with 200,000 random rows (~one percent of the full dataset), while the error bar represents its 95% CI.
(DOCX)

**S4 Fig. Sensitivity of species-level $R_{0i}$s to changes of parameter values.** Each dot represents the re-estimated $R_{0i}$ (y-axis) of a specific species (colors) after multiplying the parameter of interest (corresponding to the panel name) by a scaling factor (x-axis). The sizes of the dots represent the proportion of parameter sets that resulted in species-level seroprevalence rates within the confidence intervals of their observed values. When all parameter sets failed to match the seroprevalence rates, no dots were displayed. To assess the importance of each individual species' abundance, $\chi_i$s, instead of $\chi_{max}$, were perturbed. The definition of each parameter can be found in S2 Text.
(DOCX)

**S5 Fig. The five interventions that can lead to the largest reduction in the overall $R_0$ for each location.** X-axis represent the magnitude of the change, while y-axis represent the overall $R_0$.
(DOCX)

**S6 Fig. (A) Overall $R_0$ and (B) species-level $R_{0i}$s when different species was left out from the calibration for Survey 7.** Black bars represent the values estimated using the full dataset, while the colored bars represent the values estimated when the seroprevalence rate for one host species was left out from the calibration.
(DOCX)

## Acknowledgments

The computation is completed in the HPC Platform of Huazhong University of Science and Technology.

## Author contributions

**Conceptualization:** Qu Cheng, Junhua Tian, Banghua Chen, Zhihang Peng, Liqun Fang, Wei Liu, Yang Yang, Bethan V. Purse.

**Data curation:** Qu Cheng, Xinqiang Wang.

**Formal analysis:** Qu Cheng.

**Funding acquisition:** Qu Cheng.

**Investigation:** Qu Cheng.

**Methodology:** Qu Cheng, Yang Yang, Bethan V. Purse.

**Project administration:** Qu Cheng.

**Resources:** Qu Cheng.

**Software:** Qu Cheng, Qi Li.

**Validation:** Qi Li.

**Visualization:** Qu Cheng.

**Writing – original draft:** Qu Cheng, Xinqiang Wang.

**Writing – review & editing:** Qu Cheng, Xinqiang Wang, Qi Li, Hailan Yu, Xiaolu Wang, Chenlong Lv, Junhua Tian, Banghua Chen, Zhihang Peng, Liqun Fang, Wei Liu, Yang Yang, Bethan V. Purse.

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
