## [Decision Letter · Decision Letter 0]

Response to Reviewers
Revised Manuscript with Track Changes
Manuscript

Shaden Kamhawi

co-Editor-in-Chief

Paul Brindley

co-Editor-in-Chief

**Journal Requirements:**

At this stage, the following Authors/Authors require contributions: Xinqiang Wang, Qi Li, Hailan Yu, Xiaolu Wang, Chenlong Lv, Junhua Tian, Banghua Chen, Zhihang Peng, Liqun Fang, Wei Liu, Yang Yang, and Bethan V. Purse. Please ensure that the full contributions of each author are acknowledged in the "Add/Edit/Remove Authors" section of our submission form.

Potential Copyright Issues:

- Figure 1. Please confirm whether you drew the images / clip-art within the figure panels by hand. If you did not draw the images, please provide (a) a link to the source of the images or icons and their license / terms of use; or (b) written permission from the copyright holder to publish the images or icons under our CC BY 4.0 license. Alternatively, you may replace the images with open source alternatives. See these open source resources you may use to replace images / clip-art:

- Figure 2. Please (a) provide a direct link to the base layer of the map (i.e., the country or region border shape) and ensure this is also included in the figure legend; and (b) provide a link to the terms of use / license information for the base layer image or shapefile. We cannot publish proprietary or copyrighted maps (e.g. Google Maps, Mapquest) and the terms of use for your map base layer must be compatible with our CC BY 4.0 license.

6) Please amend your detailed Financial Disclosure statement. This is published with the article. It must therefore be completed in full sentences and contain the exact wording you wish to be published. Please ensure that the funders and grant numbers match between the Financial Disclosure field and the Funding Information tab in your submission form. Note that the funders must be provided in the same order in both places as well.

**Reviewers' comments:**

**Key Review Criteria Required for Acceptance?**

**Methods:**

-Are the objectives of the study clearly articulated with a clear testable hypothesis stated?

-Is the study design appropriate to address the stated objectives?

-Is the population clearly described and appropriate for the hypothesis being tested?

-Is the sample size sufficient to ensure adequate power to address the hypothesis being tested?

-Were correct statistical analysis used to support conclusions?

-Are there concerns about ethical or regulatory requirements being met?

Reviewer #1: (No Response)

Reviewer #2: (No Response)

**Results**

-Does the analysis presented match the analysis plan?

-Are the results clearly and completely presented?

-Are the figures (Tables, Images) of sufficient quality for clarity?

Reviewer #1: (No Response)

Reviewer #2: (No Response)

**Conclusions**

-Are the conclusions supported by the data presented?

-Are the limitations of analysis clearly described?

-Do the authors discuss how these data can be helpful to advance our understanding of the topic under study?

-Is public health relevance addressed?

Reviewer #1: 1. In the section discussing the limitations of the study, further consideration should be given to the interpretation of SFTSV-specific total IgG levels. One possible explanation for the high antibody prevalence observed in animals could be interspecies differences in antibody half-life. Additionally, repeated exposure to SFTSV may result in a boosting effect, maintaining elevated antibody titers over time.

2. The manuscript suggests that poultry may play a major role in the transmission of SFTSV. To support this claim, it would be more convincing to include a summary and discussion of research findings related to SFTSV RNA detection rates in poultry, clinical symptoms, and experimental infection studies. Since the mathematical model was based solely on existing seroprevalence data from China, it would be helpful to reference empirical studies that correlate with the modeling results derived from this "dry work."

Reviewer #2: (No Response)

**Editorial and Data Presentation Modifications?**

Reviewer #1: (No Response)

Reviewer #2: (No Response)

**Summary and General Comments**

Reviewer #1: 1. Line 42: Huaiyangshan Banyangvirus was a provisional name previously used; however, the virus has since been officially named Dabie bandavirus by the International Committee on Taxonomy of Viruses (ICTV). Therefore, I recommend deleting the former name. Additionally, since the current name was established by the ICTV in 2020, it would be appropriate to include a reference to support this information.

2. Line 46-48: The term dead-end hosts is currently not well-supported by sufficient evidence. References 12 and 13, which are cited in support of this term, are not directly relevant to dead-end host dynamics. This sentence should be restructured accordingly to avoid potentially misleading interpretations

3. The analysis in the manuscript is based on seroprevalence data, primarily derived from ELISA-based studies. It should be noted that ELISA assays using SFTSV-NP as the antigen have been reported to cross-react with other bunyaviruses, as shown in previous Chinese studies. To ensure accurate information delivery to readers, a brief summary of the methodologies used to determine seropositivity should be included.

Reviewer #2: This manuscript presents exploration of a multihost mathematical model aimed at evaluating the role different tick host species play in the transmission of SFTS. The work combines species-specific seroprevalence estimates from across a number of reported surveys, with a simple but appropriate mathematical model, and a sensible approach to calibration, to identify the key hosts the drive transmission (contribute to total R0) in each survey, and implications for different targeted control strategies.

Overall the work is well presented, logical, and leads to some interesting insights – in particular highlighting the potential important role that poultry play in dominating transmission in many of the sites, but also showing how identity of the key host species can vary between locations. It also serves as a neat illustration of how limited data and simple mathematical models can be combined to reveal important insights, and generate new hypotheses about the roles different host species play in such complex systems. As such I think this paper has the potential to make a good contribution to the wider field. I do though have several points that I would like to see the authors address, primarily related to improving clarity in presentation and interpretation:

1. Line 147 – the simulated seroprevalence is given as the equilibrium ‘recovered’ (Ri) size of each host species. Potentially though, actively infected hosts (the Ii classes) would also test seropositive. I understand those may be quite short durations before recovery, but it may alter results if simulated seroprevalence is given as Ii+Ri. Have the authors assessed sensitivity of their results to this?

More generally, it would be useful to have a little more information about the calibration process here. I found myself wondering how closely the simulated values had to ‘match’ the observed (I later saw in the Supp Info that they just had to fall within the 95% CIs of the observed prevalence values). Also it wasn’t immediately clear how parameter values were generated – again it is stated in the Supp Info that these were drawn randomly etc – but a little more detail in the main text will help the reader understand exactly what was done.

2. Line 178 – it isn’t clear why acaricide usage would reduce contact rates, instead of (or in addition to) increasing tick mortality. Is the assumption that acaricides provide a repellent protective effect to treated hosts?

3. Figure 2 – panel B gives a nice visual representation of species-specific prevalences, but these in themselves are of limited value as indicators of contributions to transmission, as host abundance is very important (as shown in the expressions for the R0i, and subsequent results) – it is the number of infecteds that drive transmission. Could equivalent plots of Figure 2B be provided that show estimated host abundance across the different surveys, or even calculated numbers infected (seroprevalence x host abundance)?

4. Figure 3 gives a nice representation of the estimated relationships between Overall R0 and species-specific seroprevalences – however those are then used to infer that goats/sheep, cattle, poultry, dogs, hedgehogs “play a [more] important role…in sustaining transmission” (lines 244-245). But I think it’s very hard to infer cause-and-effect from relationships like this. It’s not impossible (confirmed by the later analyses) that for at least some of these species, they may act as ‘sentinels’, serving as an indicator of high transmission (ie, they show high seroprevalence when R0 is high) – but don’t necessarily play much of a role in driving high R0. Some care or extra nuance in interpreting these relationships may be useful.

5. ‘Impact of missing key host species’ – I understand the use of Survey 9 to explore the consequences of missing certain species, given it was the one with the greatest number surveyed – but any results from that are going to depend on the specific findings from that original survey. If, for example, Survey 7 was used (which had the next highest number of host species surveyed, and found that goats/sheep were the dominant transmission hosts) we would presumably see quite different outcomes. I think it’s nice for the authors to have carried out this assessment, but it should be made clear that the specific results depend strongly on the initial findings relating to Survey 9 specifically.

And relating to this analysis, I understand why the R0i for the ‘unremoved’ species increase when poultry are removed (the remaining species need to collectively contribute more to reach the same overall R0 without poultry) – but I don’t see why that overall R0 should increase when poultry are removed. I would imagine (if I’ve understood the process) that should stay approximately fixed – but I assume I’m missing something here. Some explanation of this phenomenon would be very useful.

And a couple of minor typos:

- Line 173, change “was” to “were”

- Lines 367-68 – should probably read: “…though uptake will depend on these fit into…”

PLOS authors have the option to publish the peer review history of their article (what does this mean? ). If published, this will include your full peer review and any attached files.

**Do you want your identity to be public for this peer review?** For information about this choice, including consent withdrawal, please see our Privacy Policy .

Reviewer #1: No

Reviewer #2: No

**Figure resubmission:****Reproducibility:** To enhance the reproducibility of your results, we recommend that authors of applicable studies deposit laboratory protocols in protocols.io, where a protocol can be assigned its own identifier (DOI) such that it can be cited independently in the future. Additionally, PLOS ONE offers an option to publish peer-reviewed clinical study protocols. Read more information on sharing protocols at https://plos.org/protocols?utm_medium=editorial-email&utm_source=authorletters&utm_campaign=protocols

---

## [Decision Letter · Decision Letter 1]

Dear Dr. Cheng,

We are pleased to inform you that your manuscript 'Contributions of different host species to the natural transmission of severe fever with thrombocytopenia syndrome virus in China' has been provisionally accepted for publication in PLOS Neglected Tropical Diseases.

Best regards,

Nam-Hyuk Cho

Academic Editor

David Safronetz

Section Editor

Shaden Kamhawi

co-Editor-in-Chief

Paul Brindley

co-Editor-in-Chief

Reviewer's Responses to Questions

**Key Review Criteria Required for Acceptance?**

**Methods**

-Are the objectives of the study clearly articulated with a clear testable hypothesis stated?

-Is the study design appropriate to address the stated objectives?

-Is the population clearly described and appropriate for the hypothesis being tested?

-Is the sample size sufficient to ensure adequate power to address the hypothesis being tested?

-Were correct statistical analysis used to support conclusions?

-Are there concerns about ethical or regulatory requirements being met?

Reviewer #1: (No Response)

Reviewer #2: (No Response)

**Results**

-Does the analysis presented match the analysis plan?

-Are the results clearly and completely presented?

-Are the figures (Tables, Images) of sufficient quality for clarity?

Reviewer #1: (No Response)

Reviewer #2: (No Response)

**Conclusions**

-Are the conclusions supported by the data presented?

-Are the limitations of analysis clearly described?

-Do the authors discuss how these data can be helpful to advance our understanding of the topic under study?

-Is public health relevance addressed?

Reviewer #1: (No Response)

Reviewer #2: (No Response)

**Editorial and Data Presentation Modifications?**

Reviewer #1: (No Response)

Reviewer #2: (No Response)

**Summary and General Comments**

Reviewer #1: Overall, the authors answered the questions well.

Reviewer #2: The authors have done a good job addressing my earlier comments - I appreciate the additional analyses they conducted to explore senstitivity to including 'infecteds' in the calculation of seropositivity (I don't feel strongly whether this should be included in the Supplementary Materials - but perhaps it would be useful to include a brief mention that it was explored, with no effect on estimated values or conclusions). I also appreciate the additional exploration of Survey 7 data as the basis for the 'missing species' analyses. Overall I am happy with the revisions made, and have no further comments.

PLOS authors have the option to publish the peer review history of their article (what does this mean? ). If published, this will include your full peer review and any attached files.

**Do you want your identity to be public for this peer review?** For information about this choice, including consent withdrawal, please see our Privacy Policy .

Reviewer #1: No

Reviewer #2: No

---

## [Editor Report · Acceptance letter]

Dear Dr. Cheng,

We are delighted to inform you that your manuscript, "Contributions of different host species to the natural transmission of severe fever with thrombocytopenia syndrome virus in China," has been formally accepted for publication in PLOS Neglected Tropical Diseases.

Best regards,

Shaden Kamhawi

co-Editor-in-Chief

Paul Brindley

co-Editor-in-Chief
